# Novel Links between TORC1 and Traditional Non-Coding RNA, tRNA

**DOI:** 10.3390/genes11090956

**Published:** 2020-08-19

**Authors:** Yoko Otsubo, Yoshiaki Kamada, Akira Yamashita

**Affiliations:** 1National Institute for Basic Biology, Nishigonaka 38, Myodaiji, Okazaki, Aichi 444-8585, Japan; otsubo@nibb.ac.jp (Y.O.); yoshikam@nibb.ac.jp (Y.K.); 2National Institute for Fusion Science, 322-6 Oroshi, Toki, Gifu 509-5292, Japan; 3Center for Novel Science Initiatives, National Institutes of Natural Sciences, Nishigonaka 38, Myodaiji, Okazaki, Aichi 444-8585, Japan; 4Department of Basic Biology, School of Life Science, SOKENDAI (The Graduate University for Advanced Studies), Nishigonaka 38, Myodaiji, Okazaki, Aichi 444-8585, Japan

**Keywords:** TORC1, tRNA, budding yeast, fission yeast

## Abstract

Target of rapamycin (TOR) is a serine/threonine kinase that modulates cell growth and metabolism in response to environmental changes. Transfer RNA (tRNA) is an abundant and ubiquitous small non-coding RNA that is essential in the translation of mRNAs. Beyond its canonical role, it has been revealed that tRNAs have more diverse functions. TOR complex 1 (TORC1), which is one of the two TOR complexes, regulates tRNA synthesis by controlling RNA polymerase III. In addition to tRNA synthesis regulation, recent studies have revealed hidden connections between TORC1 and tRNA, which are both essential players in eukaryotic cellular activities. Here, we review the accumulating findings on the regulatory links between TORC1 and tRNA—particularly those links in the budding yeast *Saccharomyces cerevisiae* and the fission yeast *Schizosaccharomyces pombe*.

## 1. TORC1 and tRNA

Target of rapamycin (TOR) is a highly conserved serine/threonine kinase among eukaryotes. TOR acts as a master regulator that controls cell growth and metabolism in response to environmental changes (reviewed in [1,2]). TOR forms two distinct multi-protein complexes—TOR complex 1 (TORC1), and complex 2 (TORC2)—which regulate a variety of cellular activities. Between these, only TORC1 is sensitive to rapamycin. The activity of TORC1 is regulated by nutrients, growth factors, and cellular energy. In mammals, growth factors and cellular energy stimulate the activity of mTORC1 through Rheb GTPase, and via the inhibition of the tuberos sclerosis complex (TSC), which functions as a GTPase-activating protein for Rheb. When responding to the availability of amino acids, mTORC1 is activated via RAG GTPases in a TSC-independent pathway. Under nutrient-rich conditions, TORC1 promotes anabolic processes, such as protein, nucleotide, and lipid synthesis, while inhibiting catabolic processes, such as autophagy.

Transfer RNAs (tRNAs) are abundant, small, and ubiquitous non-coding RNAs that are essential for decoding genetic information, achieving this by delivering amino acids to the growing polypeptide chains in ribosomes. Each tRNA is charged with a cognate amino acid on its 3′ end, in an ATP-dependent manner. This process is catalyzed by aminoacyl-tRNA synthetases (ARSs). Each aminoacyl-tRNA is brought to the A-site of the ribosome by the translation elongation factor 1A; then, its anticodon forms base pairs with a corresponding mRNA codon. In recent years, a growing number of reports have shown that tRNAs have a wide range of cellular functions beyond their canonical role as adaptor molecules during protein synthesis [3,4,5]. For example, tRNA-derived small RNAs have been extensively studied and demonstrated to be novel, functional, non-coding RNAs (reviewed in [6,7]). tRNA is an essential biomolecule during translation, and TORC1 participates in the regulation of translation, suggesting a promising relationship between tRNA and TORC1. In fact, accumulating reports have demonstrated the involvement of tRNA, and its related factors, in the TORC1 pathway. In this review, we highlight recent findings on the TORC1-related roles of tRNA, and its related factors, in the budding yeast *Saccharomyces cerevisiae* and the fission yeast *Schizosaccharomyces pombe*, both of which have contributed to our understanding of the cellular signaling pathway, including TOR signaling.

## 2. tRNA-Mediated Inactivation of TORC1 in *S. cerevisiae*

In mammalian cells, the Rag complex, which is composed of RagA/RagB and RagC/RagD, regulates TORC1 by responding to amino acid levels. Rag proteins are conserved in *S. cerevisiae*, and the counterparts of RagA/RagB and RagC/RagD are Gtr1 and Gtr2, respectively [8,9,10]. The *S. cerevisiae* genes that encode the Rag proteins, namely *GTR1* and *GTR2*, are nonessential; this suggests that the Rag system is not the sole pathway to amino acid sensing by TORC1. In a study wherein the alternative mechanisms underlying the TORC1 regulation were elucidated, the tRNA-mediated inactivation of TORC1 in *S. cerevisiae* was demonstrated [11].

tRNA severely inhibits in vitro TORC1 activity. The inactivation of RNA polymerase III, which transcribes tRNA, results in the maintenance of TORC1 activity upon starvation. From these observations, it has been proposed that amino acid starvation increases the number of free uncharged tRNA that is liberated from protein synthesis; then, the accumulated free tRNA inactivates TORC1 (Figure 1). The result, that TORC1 activity is impaired in ARS mutants, supports this attractive model. tRNA may be the best player to monitor amino acid levels as it enables a response to the shortage in all kinds of amino acids.

## 3. tRNA Precursor-Mediated TORC1 Regulation in *S. pombe*

The synthesis of tRNAs involves several steps of post-transcriptional processing. These events include the removal of the 5′ leader, processing of the 3′ trailer, splicing of introns, addition of the 3′-terminal CCA residues, and modification of multiple nucleoside residues. In *S. pombe*, TORC1 represses sexual differentiation and exerts an essential function for vegetative growth, via a nutrient-sensing pathway [12,13,14,15,16], while TORC2 is required for sexual differentiation [17,18,19]. The temperature-sensitive mutants of *tor2*, which encodes the catalytic subunit of *S. pombe* TORC1, induce an ectopic initiation of sexual differentiation, even under nutrient-rich conditions at non-permissive temperatures. Several mutants that phenocopy the TORC1 mutant have been isolated. Intriguingly, many of their responsible genes encode factors that are involved in tRNA expression and modification, such as ARSs, a subunit of RNA polymerase III, or a tRNA specific adenosine deaminase subunit [20]. TORC1 activity is downregulated in these tRNA-related mutants; this suggests that tRNA may be involved in TORC1 regulation in *S. pombe*. The observation that the overexpression of tRNA precursors prevents TORC1 downregulation upon nutrient starvation and inhibits sexual differentiation indicates that tRNA precursors regulate TORC1 activity in a positive manner (Figure 2). Consistently, the expression of tRNA precursors drastically decreases with nutrient depletion. Moreover, nutrient starvation does not significantly affect the level of aminoacylation and the total amount of tRNA [20]. Although precise mechanisms on how tRNA precursors regulate the TORC1 activity remain unknown, Gtr1, a counterpart of RagA/B, is dispensable for tRNA precursor-mediated TORC1 regulation [20]. Notably, *S. pombe* Rag GTPases negatively regulate TORC1 activity in contrast to the mammalian Rag complex [21,22].

## 4. tRNA Modification and TORC1

Rapamycin has no impact on the cell growth of *S. pombe* wild-type cells, whereas mutants with reduced TORC1 activity are sensitive to rapamycin [13,23,24]. In a genetic screening for rapamycin sensitive mutants, the genes involved in tRNA modification were identified as those with the highest enrichment [25] (Table 1), thereby suggesting a close relationship between tRNA modification and TORC1. Among all types of cellular RNAs, tRNAs undergo highly diverse post-transcriptional modifications [26,27]. tRNA modifications are vital for the folding, stability, and function of tRNAs [5,28,29,30]; these modifications are found on approximately 12% of the residues of all sequenced tRNAs from a wide range of organisms [31]. Furthermore, more than 100 tRNA modifications have been discovered according to the MODOMICS database (http://genesilico.pl/modomics/) [32].

tRNA modifications occur most frequently within the anticodon loop, especially at positions 34 and 37. Position 34, which is the so-called wobble position, pairs with a third mRNA codon base, in the A-site of the ribosome, during translation [33]. A uridine base at the wobble position, U34, is carboxymethylated by the highly conserved Elongator complex [30,34,35,36,37].

Interestingly, in *S. pombe*, mutant strains that lack the Elongator complex subunits show exaggerated sensitivity to rapamycin [25,38] (Table 1). The deletion of the genes that encode the Elongator subunits results in the lower expression of TORC1 repressors, such as Tsc1 and Tsc2 [39]. Furthermore, the overexpression of *tor2*, which encodes the catalytic subunit of *S. pombe* TORC1, increases the sensitivity to rapamycin. This observation suggests that the hyperactivation of TORC1 leads to a higher sensitivity to rapamycin, as in the case of the downregulation of TORC1 activity, even though there may be a possibility that the overexpression of the catalytic subunits results in the downregulation of TORC1 by perturbing the balance of the components. From these observations, it is concluded that Elongator contributes to the downregulation of TORC1 by promoting the expression of repressors, such as Tsc1 and Tsc2. The positive regulation of TORC2 by Elongator has also been proposed [39].

The participation of tRNA modification in TORC1 regulation has been found in *S. cerevisiae*; however, its effect on TORC1 is opposite in both yeasts. In *S. cerevisiae*, the Elongator mutant cells have been shown to exhibit hypersensitivity to rapamycin [40,41,42]. Moreover, in the tRNA anticodon loop modification mutants (e.g., a double mutant of modification factors at positions 34 and 37), starvation responses, such as the starvation–responsive gene expression and autophagy, which are prevented by TORC1, are untimely observed under nutrient-rich conditions [43]. This implies that tRNA modification acts positively on TORC1 (Figure 3). While many of the TORC1 regulators are conserved in *S. cerevisiae* and *S. pombe*, there are differences. For instance, *S. cerevisiae* does not have homologs of TSC1 and TSC2. Thus, it is rational that the roles of tRNA modification in TORC1 signaling differ between two yeast species.

Rapamycin cannot prevent the growth of *S. pombe* [23]. However, the simultaneous treatment of rapamycin and caffeine, the latter of which is also known to decrease TORC1 activity [44,45], inhibits *S. pombe* growth [46]. Gene expression profiling was performed, after the combined treatment of rapamycin and caffeine, using DNA microarrays [47]. We surveyed the genes that are upregulated or downregulated by more than three-fold after treatment with rapamycin and caffeine, and selected the genes related to the tRNA metabolic pathway or the tRNA binding in an *S. pombe* database, PomBase (https://www.pombase.org). Shown in Table 2 and Table 3 are the tRNA-related genes that are upregulated or downregulated, via the inactivation of TORC1. Further studies of these genes would shed light on the roles of tRNA-related factors in the TORC1 pathway.

## 5. tRNA Nuclear Transport and TORC1

As mentioned above, tRNAs are synthesized as tRNA precursors (pre-tRNAs) prior to the post-transcriptional modifications. In *S. cerevisiae*, intron-containing pre-tRNAs are exported from the nucleus to the cytoplasm for the removal of introns, and the spliced tRNAs return to the nucleus via a so-called tRNA retrograde transport [48,49]. Amino acid starvation induces a nuclear accumulation of spliced tRNAs [48]; this suggests the involvement of TORC1 in tRNA retrograde transport. Two independent studies have demonstrated the link between TORC1 and tRNA retrograde transport [50,51]; however, the detailed mechanisms are yet to be addressed. Whitney et al. showed that rapamycin treatment during amino acid starvation prevents the nuclear accumulation of spliced tRNA, whereas the treatment with rapamycin under nutrient-rich conditions does not induce the nuclear tRNA accumulation [50]. This suggests that TORC1 plays a crucial role in tRNA accumulation during amino acid starvation. Pierce et al. demonstrated that the inhibition of TORC1 by rapamycin results in the nuclear accumulation of spliced tRNAs even in nutrient-rich conditions, thus implying that TORC1 regulates the nuclear re-export of retrograde transported spliced tRNAs [51]. Therefore, the role of TORC1 in tRNA localization must be clarified in further studies.

Notably, the *S. pombe* mutant cells that were lacking Los1, which is a nuclear tRNA export receptor, showed sensitivity to rapamycin [25] (Table 1). There may be a relationship between TORC1 and the regulation of the tRNA localization in *S. pombe*.

## 6. tRNA Synthesis by RNA Polymerase III and TORC1

Eukaryotes have three major RNA polymerases: Pol I, II, and III. tRNAs are synthesized by Pol III [52,53]. Pol III activity is regulated in response to diverse extracellular signals. Moreover, the rapid repression of RNA pol III-dependent transcription ensures cellular survival against environmental stress. TORC1 regulates pol III transcription by controlling Maf1, which is the evolutionarily conserved repressor of pol III. Here, we present a brief overview of the TORC1-mediated regulation of Maf1. For more details on the function and regulation of Maf1, please refer to other excellent reviews [54,55,56,57].

Maf1 represses pol III-dependent transcription during the various stress conditions, including nutrient starvation, rapamycin treatment, and DNA damage [58,59]. Maf1 is phosphorylated during normal growth conditions. Under stress conditions, Maf1 is dephosphorylated; this leads to the nuclear import of Maf1. Subsequently, Maf1 in the nucleus binds to pol III and represses pol III activity [60,61,62]. The phosphorylation of Maf1 is largely mediated by the TORC1 pathway. While mammalian Maf1 is phosphorylated by mTORC1, *S. cerevisiae* Maf1 is mainly phosphorylated by the Sch9 kinase, which is a target of TORC1 [63,64,65,66,67]. The direct phosphorylation of *S. cerevisiae* Maf1 by TORC1 has been observed, at least in vitro [68]. In *S. pombe*, the phosphorylation of Maf1 has been reported to be dependent on TORC1; however, whether TORC1 is directly involved or not remains unclear [69].

In *S. cerevisiae*, TORC1 also regulates tRNA synthesis through the LAMMER/Cdc-like kinase Kns1 and casein kinase II CK2 [70]. Kns1 is negatively regulated downstream of TORC1. When TORC1 is inhibited, the activated Kns1 phosphorylates Ckb1, which is a CK2 regulatory subunit. This causes a reduction in the CK2 occupancy of the tRNA genes, and results in the repression of pol III. Kns1 also phosphorylates Rpc53, a pol III subunit [71]; the significance of this phosphorylation remains unknown.

In *S. cerevisiae*, several subunits of pol III, including Rpc82, Rpc53, and Ret1, are sumoylated in a TORC1-dependent manner. The sumoylation of Rpc82 contributes to the stabilization of the pol III complex and is required for an efficient tRNA transcription under optimal growth conditions [72].

A recent study has shown an interesting relationship between pol III and the longevity downstream of TORC1 in *S. cerevisiae*, the fly, and nematode [73]. A reduction in pol III activity has been reported to result in the expansion of the chronological lifespan of *S. cerevisiae*. In *S. pombe*, the downregulation of TORC1 upon nutrient starvation causes the activation of the GATA transcription factor Gaf1 [74]. Recently, Gaf1 has been shown to bind tRNA genes and repress their transcription, thereby leading to the extension of the chronological lifespan [75].

## 7. Leucyl-tRNA Synthetase and TORC1

In mammals and *S. cerevisiae*, the heterotrimeric protein complexes GATOR1 and SEACIT function as GTPase-activating proteins for RagA/RagB and Gtr1, respectively. GATOR2 and SEACAT bind to and negatively regulate GATOR1 and SEACIT. In mammals, Sestrin 2 binds to leucine and regulates the GATOR2–GATOR1 pathway. In *S. cerevisiae*, leucine activates TORC1 via Gtr1; however, there is no Sestrin ortholog [76].

In addition to Sestrin 2, *S. cerevisiae* and mammalian leucyl-tRNA synthetase (LeuRS) have been reported to act as a cytoplasmic leucine sensor to activate TORC1, although the detailed mechanisms on how LeuRS acts in the TORC1 pathway are different between these two species [77,78,79,80].

In *S. cerevisiae*, LeuRS Cdc60 was isolated as a coprecipitating protein with Gtr1 [80]. LeuRS has two functionally separate activities: an essential aminoacylation activity and nonessential amino acid editing activity. A model has been proposed that leucine-bound LeuRS interacts with Gtr1 through the editing domain and positively regulates TORC1.

While the novel function of LeuRS as a leucine sensor is appealing, we have proposed alternative rational models for both *S. pombe* and *S. cerevisiae*. We isolated *S. pombe* ARS mutants, including a LeuRS mutant, during a screening for mutants that phenocopy the TORC1 mutant [20]. In the mutant cells, the expression of tRNA precursors decreases, thereby suggesting a monitoring system that affects the expression of tRNA precursors by checking the subsequent aminoacylation step. We considered that the reduction in tRNA precursors might have caused the downregulation of TORC1, as mentioned above. In *S. cerevisiae*, in addition to LeuRS, other ARSs, such as HisRS and IleRS, are involved in TORC1 regulation [11]. Therefore, we have proposed the model that uncharged tRNA accumulates in *S. cerevisiae* ARS mutant cells, and that the accumulated free tRNAs inhibit TORC1 activity. Further studies would reveal similarities and differences in the mode of action of tRNAs in the TORC1 pathway of the two yeast species.

## 8. Conclusions

There is accumulating evidence that tRNAs have more diverse functions than originally thought. For instance, tRNA-derived small RNAs (tsRNAs) have drawn increasing attention; tsRNAs are produced by cleavage at specific sites in tRNAs or pre-tRNAs, and have various biological functions, including the regulation of gene expression, inhibition of translation, prevention of apoptosis, and regulation of epigenetic inheritance. Furthermore, tsRNAs have been demonstrated to be involved in tumorigenesis [3,6,7,81]. However, their detailed mechanisms have not been fully understood.

It has long been known that TORC1 regulates tRNA synthesis in downstream events. As we have discussed here, studies over the past few years have uncovered the involvement of tRNA in the regulation of TORC1 activity, in both *S. cerevisiae* and *S. pombe*. tRNA is likely to be one of the ideal molecules that transduces nutrient availability, especially for amino acid availability, to TORC1, as it enables the fine-tuned regulation of the TORC1 activity in response to the cellular levels of each of the 20 amino acids. Although, as discussed above, the modes of action of tRNA seem to be different between *S. cerevisiae* and *S. pombe*, it is intriguing that tRNA plays important roles in these two evolutionarily distant yeast species. A comparison between the two divergent systems will bring us closer to understanding the novel regulatory mechanisms of TORC1 signaling. Further investigations, not only in yeasts but also in other eukaryotes, would lead us to a comprehensive understanding of the promising tRNA roles in the TORC1 pathway.

## Figures and Tables

**Figure 1 genes-11-00956-f001:**
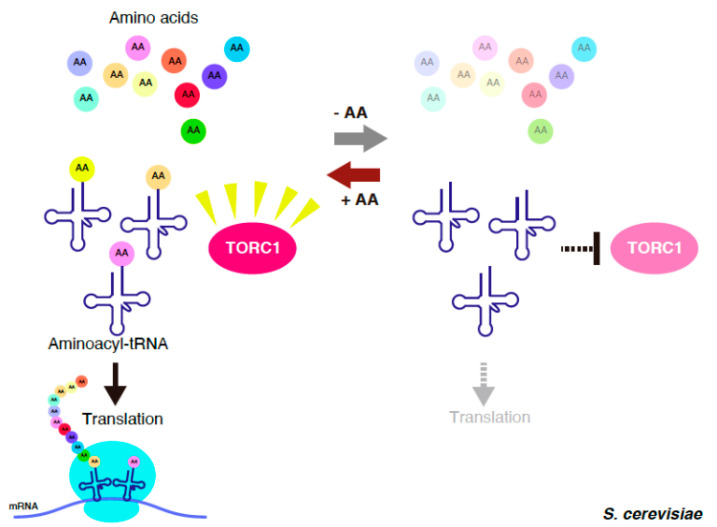
Transfer RNA (tRNA)-mediated target of rapamycin complex 1 (TORC1) regulation in *Saccharomyces cerevisiae*. Under amino acid-rich conditions, tRNAs are charged with cognate amino acids, and the resultant aminoacyl-tRNAs are delivered to the ribosome for translation. Upon amino acid starvation, free uncharged tRNAs increase, and the accumulated free tRNAs inactivate TORC1.

**Figure 2 genes-11-00956-f002:**
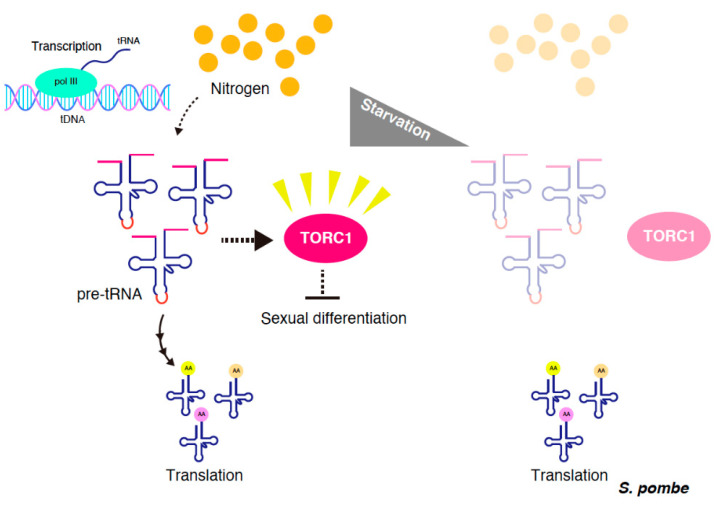
tRNA precursor-mediated TORC1 regulation in *Schizosaccharomyces pombe*. Under nitrogen-rich conditions, tRNA precursors positively regulate TORC1 activity. Under nitrogen-starved conditions, the expression of tRNA precursors decreases, resulting in the inactivation of TORC1.

**Figure 3 genes-11-00956-f003:**
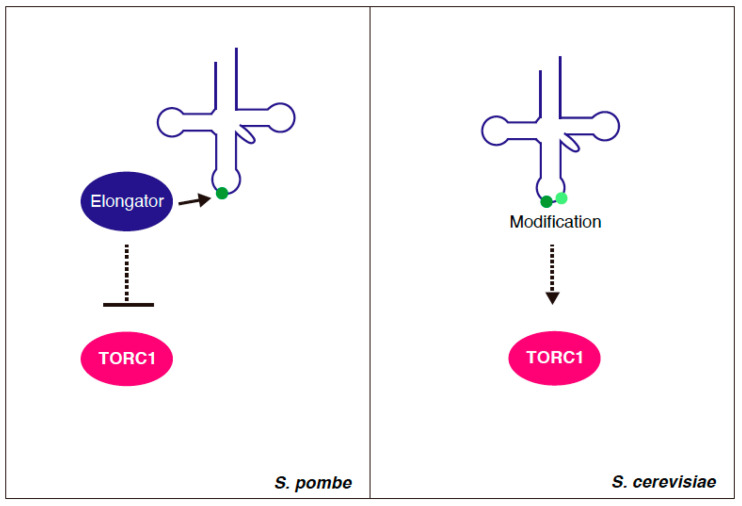
tRNA modification and TORC1. In *Schizosaccharomyces pombe*, Elongator, which catalyzes modification of the wobble base, negatively regulates TORC1. In *Saccharomyces cerevisiae*, tRNA modification acts positively on TORC1.

**Table 1 genes-11-00956-t001:** tRNA-modification genes whose deletion causes rapamycin sensitivity in *S. pombe*.

Systematic ID	Gene Name	Product
SPAC30C2.04	*asc1*	Cofactor for cytoplasmic methionyl-and glutamyl-tRNA synthetases
SPBC2G5.03	*ctu1*	Cytosolic thiouridylase subunit
SPAC25B8.05	*deg1*	tRNA-pseudouridine synthase
SPBC36.07	*elp1*	Elongator complex WD repeat protein
SPAC29A4.20	*elp3*	Elongator complex tRNA uridine (34) acetyltransferase subunit
SPCC11E10.06c	*elp4*	Elongator complex subunit
SPBC3H7.10	*elp6*	Elongator complex subunit
SPAC30.02c	*kti12*	Elongator complex associated protein
SPAC57A10.10c	*sla1*	La protein, tRNA chaperone
SPBP8B7.09c	*los1*	Karyopherin/importin-β family nuclear import receptor

**Table 2 genes-11-00956-t002:** tRNA-related *S. pombe* genes upregulated more than three-fold after treatment with rapamycin and caffeine.

Systematic ID	Gene Name	Product	Viability of Deletion Mutant
SPBC16D10.10	*tad2*	tRNA specific adenosine deaminase subunit	inviable
SPCC4B3.01	*tum1*	Thiosulfate sulfurtransferase, involved in tRNA wobble position thiolation	unknown

**Table 3 genes-11-00956-t003:** tRNA-related *S. pombe* genes downregulated more than three-fold after treatment with rapamycin and caffeine.

Systematic ID	Gene Name	Product	Viability of Deletion Mutant
SPAC9G1.12	*cpd1*	tRNA (m1A) methyltransferase complex catalytic subunit	viable
SPAC20G8.09c	*nat10*	rRNA/tRNA cytidine N-acetyltransferase	depends on conditions
SPCC126.03	*pus1*	TruA family tRNA/U2 snRNA pseudouridine synthase	viable
SPAC22A12.05	*rpc11*	DNA-directed RNA polymerase III complex subunit	inviable
SPAC57A10.10c	*sla1*	La protein, tRNA chaperone	viable
SPBC16D10.02	*trm11*	tRNA (guanine-N2-)-methyltransferase catalytic subunit	viable
SPAC31A2.02	*trm112*	eRF1 methyltransferase complex and tRNA (m2G10) methyltransferase complex regulatory subunit	viable
SPCPB16A4.04c	*trm8*	tRNA (guanine-N7-)-methyltransferase catalytic subunit	viable
SPCC18.13	*trm82*	tRNA (guanine-N7-)-methyltransferase WD repeat subunit	viable

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
