# Peer review of "Novel Links between TORC1 and Traditional Non-Coding RNA, tRNA"

_genes, 2020, doi:10.3390/genes11090956_

Round 1

Reviewer 1 Report

Otsubo and colleagues propose a very timely, concise and interesting review on the connection between TORC1 and tRNA. The topic is complex and they succeed in providing a clear summary of what is known, mostly in yeast models, and the open questions. They discuss their recent elegant work but this is well-balanced with contributions by others.

I have only a few minor comments / suggestion.

line 114: I believe that citing the works of Anders Bystrom, whose contributions in describing Elongator as a tRNA modification complex are very extensive, is necessary. The authors may check my recent historical  perspective on this at https://www.mdpi.com/2075-4655/4/2/7

line 126: it may be worth mentioning that the TSC proteins are not present is S. cerevisiae and the fact that the regulation is different between the two yeasts may not be so surprising. There are numerous such examples and it is important that non-yeast expert fully realize that comparing the two yeast is interesting. If they would be too close, that would be much less interesting.

line 234: it is interesting that again (as for the connection to anticodon modifications) cerevisae and pombe seem to have oppositive regulation with free tRNAs inhibiting TORC1 in cerevisiae and activating TORC1 in pombe. This may be highlighted.

Beside these minor issues, to me, the review should be published without delay.

Author Response

line 114: I believe that citing the works of Anders Bystrom, whose contributions in describing Elongator as a tRNA modification complex are very extensive, is necessary. The authors may check my recent historical  perspective on this at https://www.mdpi.com/2075-4655/4/2/7

Response: We appreciate the reviewer's suggestion. Accordingly, we have cited articles by Bystrom and the comprehensive review by the reviewer (line 124).

line 126: it may be worth mentioning that the TSC proteins are not present is S. cerevisiae and the fact that the regulation is different between the two yeasts may not be so surprising. There are numerous such examples and it is important that non-yeast expert fully realize that comparing the two yeast is interesting. If they would be too close, that would be much less interesting.

Response: We would like to thank the reviewer for pointing this out. We have added sentences to explain the difference between two yeast species: "While many of the TORC1 regulators are conserved in S. cerevisiae and S. pombe, there are differences. For instance, S. cerevisiae does not have homologs of TSC1 and TSC2. Thus, it is rational that the roles of tRNA modification in TORC1 signaling differ between two yeast species" (line 141).

line 234: it is interesting that again (as for the connection to anticodon modifications) cerevisiae and pombe seem to have oppositive regulation with free tRNAs inhibiting TORC1 in cerevisiae and activating TORC1 in pombe. This may be highlighted.

Response: As per the reviewer's suggestion, we have added a following sentence: " Further studies would elucidate similarities and differences in the mode of action of tRNAs in the TORC1 pathway of two yeast species." (line 291). We have also added a following sentence in Conclusion to emphasize the importance of comparison of two distinct yeasts: "Comparison between two divergent systems will bring us closer to understand novel regulatory mechanisms of TORC1 signaling" (line 308).

Reviewer 2 Report

In this Review article, Otsubo and colleagues explore the mechanistic role that tRNAs have in regulating TORC1 signaling in budding and fission yeast, and to a lesser extent, mammals as well. How tRNAs serve as nutrient sensing and signaling effectors to activate and/or repress TORC1 (depending on the organism analyzed) remains a very poorly understood area of the TORC1 signaling field. This review does an excellent job of summarizing this limited information, as well as providing some suggestive mechanistic possibilities for the field to consider moving forward. The article is very well written, and I believe it to be suitable for publication in Genes with the caveat that two minor changes be made which are listed below.

1) At the end of Line 79, I think it would be appropriate to cite the following article: Curr Genet

. 2001 May;39(3):166-74. doi: 10.1007/s002940100198. Fission yeast tor1 functions in response to various stresses including nitrogen starvation, high osmolarity, and high temperature. M Kawai 1, A Nakashima, M Ueno, T Ushimaru, K Aiba, H Doi, M Uritani

2) Line 198 reads “Kns1 are…”. This should be changed to “Kns1 is…”.

Author Response

1) At the end of Line 79, I think it would be appropriate to cite the following article: Curr Genet. 2001 May;39(3):166-74. doi: 10.1007/s002940100198. Fission yeast tor1 functions in response to various stresses including nitrogen starvation, high osmolarity, and high temperature. M Kawai 1, A Nakashima, M Ueno, T Ushimaru, K Aiba, H Doi, M Uritani

Response: In the article by Kawai et al., the tor2 gene, which encodes a catalytic subunit of TORC1, was identified through a database search. However, the authors focused on the analysis of the other TOR gene, tor1, whose gene product constitutes TORC2. We have modified the text and added the citation: "In S. pombe, TORC1 represses sexual differentiation and exerts an essential function for vegetative growth via a nutrient-sensing pathway, while TORC2 is required for sexual differentiation" (line 80).

2) Line 198 reads “Kns1 are…”. This should be changed to “Kns1 is…”.

Response: We apologize for the mistake. We have corrected the text (line 236).